# The Impact of Plant-Based Dietary Patterns on Cancer-Related Outcomes: A Rapid Review and Meta-Analysis

**DOI:** 10.3390/nu12072010

**Published:** 2020-07-06

**Authors:** Esther Molina-Montes, Elena Salamanca-Fernández, Belén Garcia-Villanova, Maria José Sánchez

**Affiliations:** 1Department of Nutrition and Food Science, University of Granada, 18014 Granada, Spain; belenv@ugr.es; 2Institute of Nutrition and Food Technology (INYTA) ‘José Mataix’, Biomedical Research Centre, University of Granada, Avenida del Conocimiento s/n, E-18071 Granada, Spain; 3Instituto de Investigación Biosanitaria ibs.GRANADA, 18012 Granada, Spain; mariajose.sanchez.easp@juntadeandalucia.es; 4CIBER of Epidemiology and Public Health (CIBERESP), 28029 Madrid, Spain; 5Andalusian School of Public Health (EASP), 18014 Granada, Spain; 6Department of Preventive Medicine and Public Health, University of Granada, 18016 Granada, Spain

**Keywords:** cancer, mortality, survival, vegan, vegetarian, Mediterranean diet, diet quality, plant-based food

## Abstract

Long-term cancer survivors represent a sizeable portion of the population. Plant-based foods may enhance the prevention of cancer-related outcomes in these patients. We aimed to synthesize the current evidence regarding the impact of plant-based dietary patterns (PBDPs) on cancer-related outcomes in the general population and in cancer survivors. Considered outcomes included overall cancer mortality, cancer-specific mortality, and cancer recurrence. A rapid review was conducted, whereby 2234 original articles related to the topic were identified via Pubmed/Medline. We selected 26 articles, which were classified into studies on PBDPs and cancer outcomes at pre-diagnosis: vegan/vegetarian diet (*N* = 5), provegetarian diet (*N* = 2), Mediterranean diet (*N* = 13), and studies considering the same at post-diagnosis (*N* = 6). Pooled estimates of the associations between the aforementioned PBDPs and the different cancer outcomes were obtained by applying random effects meta-analysis. The few studies available on the vegetarian diet failed to support its prevention potential against overall cancer mortality when compared with a non-vegetarian diet (e.g., pooled hazard ratio (HR) = 0.97; 95% confidence interval (CI): 0.88–1.06). The insufficient number of studies evaluating provegetarian index scores in relation to cancer mortality did not permit a comprehensive assessment of this association. The association between adherence to the Mediterranean diet and cancer mortality reached statistical significance (e.g., pooled HR = 0.84; 95% CI: 0.79–0.89). However, no study considered the influence of prognostic factors on the associations. In contrast, post-diagnostic studies accounted for prognostic factors when assessing the chemoprevention potential of PBDPs, but also were inconclusive due to the limited number of studies on well-defined plant-based diets. Thus, whether plant-based diets before or after a cancer diagnosis prevent negative cancer-related outcomes needs to be researched further, in order to define dietary guidelines for cancer survivors.

## 1. Introduction

The latest global cancer statistics have reported that there were 17.0 million new cases of cancer and 9.5 million cancer-related deaths (excluding non-melanoma skin cancer) in 2018 [1]. It is estimated that 43.8 million cancer patients who survived the disease for five years or more were alive in 2018, with lung, breast, colorectal, prostate, and gastric cancer contributing more prominently to this number [2,3]. Data from Europe (EU-28) are also impressive, with more than 12.1 million prevalent cancer patients in 2018 [3]. Parallel to the increase of the number of new cancer patients over the past decades, the proportion of cancer survivors has also grown. Survival rates for cancer have increased steadily for patients diagnosed with cancer from 1999 to 2007 [4]. This increased survival may be partially attributed to improved diagnosis and treatments, better access to high-quality services and earlier diagnosis [5]. However, several factors can lead to worse cancer prognosis, among which unhealthy dietary choices warrant special consideration [6].

Plant-based food (fruits, vegetables, cereals, nuts and seeds, legumes, and vegetable oils) are the main source of fiber and other bioactive compounds in the diet [7]. Particularly, plant bioactives including fiber, sulfur compounds, carotenoids, and polyphenols, present in foods such as cruciferous and allium vegetables, tomatoes, green tea, and whole grain cereal, have well-known anticarcinogenic properties [7]. Plant-based foods are therefore likely to exert anti-inflammatory and anti-oxidative effects against the development of cancer [7]. Thus, whereas an unhealthy diet is an established risk factor for several cancer types, eating plant-based foods to achieve a healthful diet has been associated with a reduced cancer risk according to the latest report on diet and cancer, released by the World Cancer Research Fund/American Institute for Cancer Research (WCRF/AICR) [8].

A considerable number of studies examined the association between individual nutrients or foods and cancer mortality [9,10]. Most of these studies have focused on associations between pre-diagnosis dietary intake and cancer-related outcomes, while only a few studies have assessed how post-diagnosis dietary intake affects these outcomes [11]. Moreover, nearly all have focused on overall cancer mortality, whereas studies on cancer-specific mortality are scarce [11,12,13,14]. In addition to the above, findings from these studies are far from conclusive. For example, inconsistent associations were reported between fruit and vegetable consumption and cancer mortality in a meta-analysis of sixteen prospective studies [15]. For other plant-based foods, including nuts [16] and legumes [9], strong associations with cancer mortality have been observed. In contrast, olive oil consumption was not found to be associated with cancer mortality in several studies [17,18]. Therefore, until now, there is no sufficient evidence to recommend dietary factors for cancer mortality prevention.

Considering, the overall diet (i.e., the dietary pattern) is more appropriate in view of the complex nature of the diet, and the interrelation between its constituents (foods and nutrients) [19]. Dietary patterns can be assessed using the conceptual framework of a dietary pattern (a priori), or empirically (a posteriori), and allow to establish more robust associations than when considering nutrients or foods on an individual basis [20]. It is also of great importance that dietary patterns, especially a priori dietary patterns, are more suitable to derive dietary guidelines for the prevention of cancer [8,21]. However, not much attention has been paid to the influence of diet as a whole on cancer mortality outcomes. So far, the WCRF/AICR cancer prevention guidelines establish the same recommendations for cancer survivors as for the general population. Adherence to these guidelines by means of the WCRF/AICR 2007 score has been associated with a lower cancer mortality risk in a meta-analysis combining only three studies on this association [14].

Thus, whether specific dietary guidelines reduce cancer mortality risk in the population (before the diagnosis of cancer) or in cancer survivors (from post-diagnosis) remains unclear. Dietary guidelines accounting for the holistic effects of plant-based foods within a healthy diet for preventing cancer mortality are crucial in this regard. Their effects have been thoroughly examined for cancer incidence [13,22], but not with respect to cancer mortality. Hence, the aim of this review was to synthesize the current evidence regarding cancer mortality. For the purpose of this review, plant-based dietary patterns (PBDPs) were deemed to be the vegetarian diet (VD) in all its variations and the Mediterranean diet (MD). For the latter, we focused on a priori-derived MD indexes (e.g., the MD score, the alternate MD index (aMED), and others, in their various versions), whereas empirically derived a posteriori patterns were discarded. Furthermore, we distinguished between pre-diagnostic and post-diagnostic effects of PBDPs on cancer outcomes including overall cancer mortality, cancer-specific mortality, and cancer recurrence.

## 2. Materials and Methods

A rapid review methodology was chosen to conduct the review in a timely manner, following a pre-established protocol for systematic reviews (with the exception of several steps that were omitted to speed up the process, such as methodological quality assessment), and according to the general methodology of the Preferred Reporting Items of Systematic Reviews and Meta-Analyses (PRISMA) guidelines [23,24,25]. Whenever possible, results extracted from the selected studies were pooled in a meta-analysis. The steps are described in more detail below.

### 2.1. Search Strategy

Leading electronic databases (Medline via PubMed) were searched from 1999 (no studies were available before) until 20 April 2020. The search strategy used (with a combination of Mesh terms, keywords, and operators) is detailed in Appendix A. The flow diagram illustrating the process of identifying and selecting studies is shown in Appendix B.

### 2.2. Study Eligibility Criteria

The review included all original studies in humans that provided data on cancer mortality or survival in relation to PBDPs. These studies were intervention studies, cohort studies, and case-control studies. Systematic reviews or meta-analyses were not eligible, but they were included in the search to further retrieve original studies on the topic by manual search. The most updated reviews were selected as references to encompass all the available studies [11,26,27,28]. Studies that were published thereafter and those studies not included in these previous reviews were included too.

With reference to the outcomes, we considered studies the primary outcome of which was overall and/or cancer-specific mortality, and studies reporting results on other related outcomes such as recurrence and progression of the disease. Cancer survivors did not include survivors of cervical lesions or adenomas in the colon since these are considered benign or non-malignant lesions according to common histological classification of tumors. In addition, eligible studies were those evaluating dietary patterns by questionnaire assessment.

PBDPs were defined as: (i) a diet excluding meat and meat products, flesh from any animal, and seafood, or all foods of animal origin, in the strictest sense (vegetarian and vegan diet, respectively); and (ii) a diet featuring a higher consumption of fruits, vegetables, legumes, and nuts over foods of animal origin. We considered the vegetarian diet and Mediterranean diet (MD) as plant-based dietary models. For the former, vegetarian population studies or studies that ascertained compliance with the vegetarian diet standards were eligible. To establish the latter, only a priori-derived MD indexes were included due to their high translational capacity into dietary prevention guidelines. In the light of the different adaptations that have been applied to the first MD score (MDS) proposed by Trichopoulou et al. [29,30], no restrictions were applied to these indexes.

All possible comparisons between vegetarians, vegans (or both combined) with respect to omnivores or different combinations of vegetarianism (e.g., lacto-ovo vegetarians) were considered. We also considered studies that established comparisons by levels of adherence to PBDPs. Furthermore, studies assessing the association between PBDPs and cancer mortality both before and after cancer diagnosis were included.

Studies conducted in children and adolescents (aged <18 years) were excluded, as were those not reporting any risk estimate (odd ratio (OR), relative risk (RR), or hazard ratio (HR) and the corresponding 95% confidence interval (CI)) on the association between dietary patterns and the aforementioned cancer-related outcomes. Studies written in languages other than English or Spanish were also excluded.

### 2.3. Data Collection and Analysis

#### Selection of Studies

Studies were first screened by title and abstract by two reviewers (E.M.M. and E.S.F.) and the final study selection was performed based on a full text review. When there were several articles reporting results based on the same study population, we included the study reporting the most updated data. Any discrepancies were resolved by consulting a third reviewer (M.J.S.).

### 2.4. Data Extraction and Management

Data extraction was performed by three reviewers (E.M.M., E.S.F., and B.G.V.) using a predefined standardized form to collect information on: (a) study characteristics: authors and year(s), study description and design, country, study population characteristics with regard to sample size, gender, and age; (b) the assessment methods used to collect information on diet (questionnaires and tools) and cancer-related outcomes (cancer registry or medical records data), also distinguishing between pre- and post-diagnosis association studies; (c) the measured outcomes: overall cancer mortality, cancer-specific mortality, and cancer recurrence; (d) the PBDPs under consideration (vegetarian, Mediterranean, and their variations), type (population-based or a priori), and the comparison groups (e.g., high vs. low adherence or vegetarians/vegans vs. omnivores); and (e) the reported results: measures of effect size (OR, HR, and RR, with 95% CI) and confounding variables considered for adjustment.

### 2.5. Presentation of Results

The results of all studies were presented in tabular format and summarized narratively by type of dietary pattern and cancer-related outcome. To summarize these studies, we described their results and risk estimates adjusted for all potential confounders. When possible, these results were pooled in meta-analyses.

### 2.6. Meta-Analysis

We calculated the summary estimates (from log HR/RRs in cohort studies, for dichotomous outcomes and comparisons between exposure groups) and corresponding 95% CIs when at least two studies reported results on the same outcome, using random effects models to account for possible heterogeneity between studies [31].

The Cochran Q test and I2 test statistic (assuming heterogeneity if I2 > 50%) were used to assess heterogeneity between studies. Publication bias was assessed by Egger’s test and visual inspection of the funnel plots [32,33]. A p-value of < 0.05 was deemed as statistically significant. The “metafor” package in R version 3.6.3 (R software) was used for these analyses.

## 3. Results

A total of 2234 publications was identified. In addition, we considered the most up-to-date reviews addressing the association between dietary patterns and cancer mortality [11,12,13,14]. These reviews added 25 studies, not previously retrieved. Of these, 930 were excluded in the first step to retain original articles published from 1999. Then, 1256 studies were excluded on the basis of title and abstract, and 73 publications were reviewed using full text. Forty-seven publications were excluded for not fulfilling the inclusion criteria (no cancer related outcomes = 19; not about dietary patterns but single nutrient/food studies = 14; not an original study = 1; a posteriori indexes = 5; a priori but not MD = 8). Thus, 26 studies were included in this review: 5 VD-like studies [34,35,36,37,38], 12 MD-like studies [39,40,41,42,43,44,45,46,47,48,49,50,51], 2 provegetarian diet studies [52,53], and 6 post-diagnosis studies [54,55,56,57,58,59]. The study by Key et al. was not included because no risk ratios on cancer mortality were reported [60]. Results are presented in the following sections.

### 3.1. Vegetarian and Vegan Diet

With reference to cancer mortality, characteristics of the studies evaluating VDs (vs. nonvegetarians) and cancer mortality are shown in Table 1. These studies were based on large cohorts of known vegetarians and vegans also including nonvegetarians/nonvegans, who were followed-up for different outcomes including cancer mortality. Some of these cohorts were the Adventist Mortality Study, the Adventist Health Studies, the Oxford and Heidelberg Vegetarian Studies, the European Prospective Investigation into Cancer and Nutrition (EPIC) Oxford Vegetarian Study, and the Health Food Shoppers Study, which were combined in some of the selected publications [35,38,55]. Some studies based the VD assessment on questions regarding whether the participants followed a VD or not (e.g., Health Food Shoppers study); on the reported consumption of meat or fish using close-ended questions (e.g., Heidelberg study); or on dietary questionnaires aiming to identify non-consumers of meat or fish (e.g., the Adventist Health Study). The studies by Orlich et al. and Key et al. [37,61] were based on two different arms of the Adventist Health Study. Both were included to account for overall and cancer-specific (colorectal) mortality. All studies assessed the association between vegetarian eating habits and cancer mortality before cancer diagnosis (all excluded prevalent cancer cases at recruitment), and all assessed cancer mortality and cause-specific mortality using national mortality registry data (death certificates).

The study by Key et al. [34] compared death rates between vegetarians and nonvegetarians and showed significant heterogeneity between the five included cohorts. This heterogeneity was probably driven by the fact that the studied populations differed with regard to socio-demographic factors and definitions of the vegetarian population groups. Risk estimates were adjusted for age, sex, and smoking, despite the fact that smoking was not collected in some of the included cohorts. Unreliable associations were reported by duration of vegetarianism due to the limited sample size. In addition, participants were asked whether they considered themselves vegetarians, but no further dietary information was collected. Two of the cohorts (the Oxford Vegetarian and Health Food Shoppers) were included in the article by Appleby et al. [35]. Later on, Appleby et al. [38] compared vegetarian vs. nonvegetarian dietary habits in relation to cancer mortality (overall and by cancer type) in some of these previous cohorts (the Oxford Vegetarian and EPIC Oxford Study from the UK). This study also combined vegans and vegetarians in the analyses but considered as a reference group, regular meat eaters (intake >5/week). Importantly, low meat eaters (intake <5/week) and fish eaters (i.e., pesco-vegetarians) were considered in separate categories. The latter study also adjusted for every possible confounding variable related to medical history and lifestyle habits. The studies by Orlich [37,61] based the assessment of vegetarian diet on the dietary information collected with a validated food frequency questionnaire (FFQ) among 73,308 participants of the Adventist Health Cohort. Using this dietary data, the authors were able to classify participants into vegans, lacto-ovo vegetarians, pesco-vegetarians, semi-vegetarians, and nonvegetarians, according to the frequency of intake of particular animal foods (eggs/dairy <1/month, fish >1/month, meats <1/week, and meats >1/week, respectively). This study was, indeed, one of the few reporting risks associated with cancer mortality comparing different types of vegetarians to nonvegetarians, as well as all kinds of vegetarians vs. nonvegetarians. No significant associations were reported with respect to cancer mortality in any of these subgroups. Finally, the study by Chang-Claude et al. [36], included 1225 vegetarians and 679 health-conscious nonvegetarians from Germany, who provided dietary information on usual frequency of consumption of plant-based foods, milk, eggs, fish, meat, and processed meat. Mortality ratios were compared for vegans and vegetarians, while RRs associated with cancer mortality and adjusted for relevant confounders were reported for vegetarians and vegans combined vs. nonvegetarians. None of these studies considered the influence of cancer treatment or other prognostic factors on the associations.

Pooled estimates of effect sizes and 95% CIs for cancer mortality comparing vegetarians vs. non-vegetarians are shown in Figure 1 and described below. Whenever possible, we removed overlapping cohorts between the studies (e.g., those included in Key et al. (1999) and Appleby (2002) [34,35]). There was no evidence for publication bias according to funnel plots and Egger tests.

#### 3.1.1. Overall Cancer Mortality

Overall cancer mortality was evaluated in four articles [35,36,37,38], with one article reporting estimates for two different cohorts [35]. The articles by Appleby [35,38] combined the study populations of the Oxford Vegetarian Study, the Health Food Shoppers Study, and the EPIC Oxford Vegetarian Study. While there seemed to be population overlap in the Oxford Vegetarian Study between the articles [35,38], it was not possible to retrieve separate risk estimates by study populations. The overall analysis was based on 139,174 participants and 1661 cancer mortality events. The mean age of study participants varied greatly between the studies (from 43 to 58 years), as well as the level of confounding adjustment: from minimal by age, sex, and smoking adjustment [35] to full adjustments [36,37,38]. Pooled estimates showed that VD (vs. nonvegetarian) was not significantly associated with overall cancer mortality (RR = 0.97; 95% CI: 0.88–1.06). There was no significant heterogeneity between the studies (I^2^ = 30%, *p* = 0.24).

#### 3.1.2. Breast Cancer Mortality

There were three articles on the association between VD and breast cancer mortality [34,35,38], which together analyzed the association within six cohorts. Potential population overlap between the Oxford Vegetarian Study cohorts could not be resolved. After pooling risk estimates, based on 228 breast cancer events among 83,985 participants, a non-significant association was observed between VD (vs. nonvegetarian) and breast cancer mortality (RR = 0.99; 95% CI: 0.67–1.44). Heterogeneity tended to be significant and became less apparent after exclusion of the Heidelberg Study included in Key et al. [34]. The association remained, however, nonsignificant (data not shown).

#### 3.1.3. Colorectal Cancer Mortality

Three studies including six cohorts reporting results for 279 colorectal cancer events among 83,985 participants, evaluated the association between VD and colorectal cancer mortality [34,35,38,61]. Vegetarian diet (vs. nonvegetarian) compliance was not significantly associated with colorectal cancer mortality after pooling risk estimates of these studies (RR = 1.03; 95% CI: 0.84–1.26). There was no heterogeneity among the studies (I^2^ = 0%, *p* = 0.54).

#### 3.1.4. Cancer Mortality for Other Cancer Types

Prostate and lung cancer mortality was evaluated in two [34,35] and three [34,35,38] articles, respectively. Both showed nonsignificant associations in pooled analyses. Gastric cancer mortality was also evaluated in two of the cohorts included in the study by Key et al. [34], but no consistent associations were reported. For other cause-specific cancer mortality, only pancreas, ovary, and lymphatic/hematopoietic cancers were evaluated in relation to VD in the study by Appleby et al. [38]. Interestingly, there was a significant inverse association of VD (vs. nonvegetarian) with pancreas cancer mortality (RR = 0.44; 95% CI: 0.26–0.76) and lymphatic/hematopoietic cancer mortality (RR = 0.47; 95% CI: 0.30–0.73), but a nonsignificant association with ovarian cancer mortality. Low meat eaters vs. regular meat eaters were also found to have lower pancreatic cancer mortality risk (RR = 0.54; 95% CI: 0.35–0.85) in this study.

### 3.2. Provegetarian Diets

Our literature search retrieved two studies on the association between provegetarian diets and cancer mortality [52,53]. Characteristics of these studies are shown in Table 2.

In the first study [52], conducted with 7216 elderly participants (mean age = 67 years) of the Spanish PREDIMED (Prevención con Dieta Mediterránea) study, the authors assessed five levels of adherence to a provegetarian diet score by considering fruits, vegetables, nuts, cereals, legumes, olive oil, and potatoes as positive components, and animal fats, eggs, fish, dairy products, and meats or meat products as negative components. As the authors showed, this score was not correlated with the traditional MD score, supporting that both scores capture different dietary dimensions of a plant-based diet. This study used dietary data obtained through a validated 137-item FFQ, which was administered yearly during the follow-up. The outcome (death) was assessed through the review of clinical records and linkage to the mortality registries. During 4.8 years of follow-up, there were 130 cancer deaths documented, evenly distributed across the levels of adherence to the score. As a consequence, a nonsignificant association with cancer mortality was observed (HR high vs. very low score = 0.66; 95% CI: 0.35–1.24). The estimated HRs were adjusted for age, gender, smoking, physical activity, educational level, energy intake, and alcohol consumption. Additional adjustment for anthropometric measures and medical history, or accounting for variations in dietary habits over time, did not change the interpretation of the results.

Three different plant-based dietary indexes were evaluated in relation to cancer mortality in the study by Baden et al. [53]. The purpose of this study was to examine whether changes in adherence to these scores in 12 years were related to cancer mortality. A total of 75,314 men and women (mean age = 63 years) from the Nurses’ Health Study (NHS) and the Health Professionals Follow-up Study (HPFS) were followed-up for cancer deaths (*N* = 4263). Dietary data were collected every four years with a validated semi-quantitative FFQ. The authors proposed an overall (standard) provegetarian plant-based score, as well as healthier and unhealthier versions of this score. These scores were based on the premise that plant-based foods have different diet quality, with foods such as fruit juices, refined grains, potatoes, sugar-sweetened beverages, and sweets and desserts presumed to be less healthy; whereas whole grains, fruits, vegetables, nuts, legumes, vegetable oils, tea, and coffee are considered the opposite. All plant-based foods were scored positively, while all other foods scored negatively. The change in adherence to the scores (at 8, 12, and 16 years) was evaluated in relation to risk of cancer death. Cancer deaths in the cohorts were identified through mortality registries and/or reported by the participant’s relatives. The authors found that an increase in adherence to the provegetarian standard score (vs. stable adherence in 12 years) was associated with a lower risk of cancer death. Concretely, 10 points increase in the adherence was associated with a 7% (95% CI: 2–11%) decrease of cancer mortality risk. No consistent associations were observed between the other provegetarian diet scores (healthy and unhealthy score) and overall cancer mortality in this study, which was an unexpected finding since decreases in consumption of healthy plant foods were associated with a higher risk of all-cause mortality. Neither were there statistically significant associations when assessing the association between 12-year change in adherence to these scores with cause-specific cancer mortality (lung, breast, and colon), except for prostate cancer (HR per 10 points increase in adherence = 0.73; 95% CI: 0.55–0.96). Adjustment variables included many potential confounders related with medical conditions, as well as body mass index (BMI).

It was not possible to meta-analyze the results of both studies due to differences in the assessment of the adherence to the plant-based score, either at baseline [52] or over time [53]. The comparison groups differed in both studies too, with regard to high vs. low adherence to the provegetarian diet score [52], and change in adherence to the score vs. stable adherence over time [53]. None of the studies accounted for cancer treatment or other prognostic factors.

### 3.3. Mediterranean Diet

Our literature search retrieved 13 studies on the association between the MD (either the original MD scores (MDS), or its derivatives including the alternate MD (aMED), the modified MD (mMSD), the MD pattern (MDP), or the relative Mediterranean diet (rMED) scores) and cancer mortality and other related-outcomes [39,40,41,42,43,44,45,46,47,48,49,50,51]. Thus, the association between adherence to the MD and cancer mortality has been examined through five different a priori derived MD indexes. All studies were cohort studies, considering validated dietary assessment tools to evaluate the adherence to the MD, and all based the outcome assessment on reliable information sources (e.g., mortality and cancer registries). All studies controlled for relevant confounders (age, sex, region, energy intake, physical activity, education, and BMI), and several also considered comorbid conditions at baseline for adjustment [40,50]. Interestingly, none of the studies on the association between MD diet and overall cancer mortality risk controlled for prognostic-related factors in the analyses. More detailed characteristics of these studies are shown in Table 3.

Among the studies analyzing the association between MDS adherence and cancer mortality risk [39,41,43,46,48], two studies showed a statistically significant protective association [39,41], but three studies did not support an association [43,46,59]. In this regard, Lassale et al. [39] studied this association in the European Prospective Investigation into Cancer and Nutrition (EPIC) study among 451,256 healthy participants, followed-up for 12.8 years. This study also evaluated adherence to rMED, and both MDS and rMED (high vs. low adherence) were associated with a statistically significant reduced overall cancer mortality risk (HR = 0.90; 95% CI: 0.88–0.92 and HR = 0.88; 95% CI: 0.86–0.90, respectively). The study by Vormund et al. [41], which included 17,861 Swiss men and women, also reported that a higher adherence to the MDS (vs. low) was associated with lower cancer mortality risk in both men and women (HR = 0.95; 95% CI: 0.92–0.99), although this association was stronger in men (HR = 0.92; 95% CI: 0.88–0.97) and absent in women. Furthermore, Lagiou et al. [43], in a cohort of 42,237 Swiss women, showed that a one-point increase in the MDS was not significantly associated with cancer mortality risk. On the other hand, a Spanish study conducted among 12,449 men and women [46], found that MDS was not significantly associated with cancer mortality risk. As for the MDP score, Cheng et al. [48] studied its association with cancer mortality risk within the prospective Iowa Women’s Health Study from the USA, which included 35,221 cancer-free women at baseline, of which 4665 died due to cancer during follow-up. The adjusted HR and 95% CI for all-cancer mortality among participants in the highest relative to the lowest quintile was 0.93 (95% CI: 0.84–1.03).

The characteristics of other MD indexes, such as the aMED [40,44,45,47,49,51], the mMED [50], and the rMED [39,42], are described in the original studies [30,63] and elsewhere [64]. In relation to their association with cancer mortality and other outcomes, the following results were reported: In general, the nine aMED studies suggested that high vs. low adherence to the aMED score was associated with a decreased cancer mortality risk [40,44,45,47,49,51]. It should be noted that HR and the corresponding 95% CIs were extracted from the figures in the study by Liese et al. [45]. This study analyzed the association within three cohorts in a standardized manner. Overall, the study showed the protective effects of the MD against cancer mortality. Two studies assessing the association with regard to the mMED score were included [50]. A study by Warensjö et al. [50] of 38,428 Swedish women, found that mMED was associated with a lower cancer mortality risk (HR high vs. low adherence = 0.81 95% CI: 0.69–0.94). Finally, the two rMED studies [39,42] showed diverging results: Lassale et al. [39] showed a statistically significant inverse association with cancer mortality risk (HR = 0.88; 95% CI: 0.86–0.90) when considering the entire EPIC study cohort, whereas Buckland et al. [42] concluded that a high compared with a low rMED score adherence was not associated with a significant reduction in mortality from overall cancer in the Spanish EPIC cohort (HR = 0.92; 95% CI: 0.75–1.12).

Pooled estimates of the effect size and 95% CIs for cancer mortality risk comparing high vs. low adherers to the MD (preferably for MDS and aMED) are depicted in Figure 2 and described below. Distinctions by type of MD score were not made. Only overall mortality could be assessed due to lack of studies on MD adherence and cancer-specific mortality (except one study on colorectal cancer mortality). There was no evidence for publication bias according to the funnel plot and Egger test (data not shown).

#### 3.3.1. Overall Cancer Mortality

We included eight studies comparing extreme groups (high vs. low) of adherence to the MD [39,45,47,48,49,50,51]. There were two studies that did not report HR by these groups [41,43]. Moreover, the study by Buckland et al. [42] was also not considered for this meta-analysis due to population overlap with the study of Lassale et al. [39]. In addition, the study by Liese contributed only with women from the Women Health Initiative Observational Study (WHI-OS) [45]. The pooled analyses (combining 74,267 cancer deaths among 1,949,146 persons) revealed that overall cancer mortality risk was significantly reduced by 16% (95% CI: 11–30). However, there was significant heterogeneity between the studies (I2 = 84%, *p* < 0.001), which disappeared after excluding the study by Cuenca-Garcia et al. [46], while risk estimates remained the same.

#### 3.3.2. Colorectal Cancer Mortality

The association between MD adherence and colorectal cancer mortality was assessed in one study [40]. In this study, conducted within the Multiethnic Cohort Study (MEC) with 4204 colorectal cancer events over follow-up and subsequent 1976 deaths (1095 were colorectal cancer-specific), a nonsignificant association was observed between high vs. low adherence to aMED and colorectal cancer mortality (HR = 0.90; 95% CI: 0.57–1.43). It is important to note that this was the only study controlling for cancer treatment variables in the analyses.

### 3.4. Post-Diagnosis Studies on Plant-Based Dietary Patterns and Cancer Outcomes

We identified six studies on this topic based on our literature search [54,55,56,57,58,59], complemented with studies included in previous reviews [11,12]. These studies were mostly focused on the MD as the reference dietary pattern, or other predefined PBDPs. Included outcomes were both cancer mortality and recurrence. A summary of the studies on post-diagnosis PBDPs and cancer mortality is shown in Table 4.

Of the selected studies, there was only one intervention study [57]. This study was conducted within the Women’s Healthy Eating and Living (WHEL) Study, and included 3088 females with non-metastatic breast cancer, who were randomized into an intervention and control group. The intervention group received advice through telephone and cooking classes on how to adopt the plant-based diet defined as a low-fat and high-fiber diet, characterized by a daily intake of five vegetable servings, two glasses of vegetable juice, three fruit servings, 30 g of fiber, and 15% to 20% of energy intake from fat. The control group received advice on maintaining a healthy diet only. Compliance with the intervention was controlled through dietary records during the intervention phase, which lasted six months. Consideration was given to potential confounding of the association by prognostic factors of the disease, such as stage, and dietary habits at baseline. Results of this study did not support that a plant-based diet is associated with a reduced risk of cancer death or cancer recurrence. Although this was the only study available, the long follow-up, large sample size, low residual confounding risk, and the adequacy of procedures during the intervention and follow-up, makes the obtained results highly robust.

The remaining studies were cohort studies. Two of them were conducted within the NHS [54,55]. Both assessed the association between adherence to the MD by means of the alternate MD score (aMED) and cancer mortality with regard to breast cancer [55] or colorectal cancer [54]. Early-stage breast (*N* = 2729) and colorectal cancer (*N* = 1201) patients in these studies were females, who were diagnosed with the disease during the follow-up of the cohort and were followed-up thereafter for all-cause mortality and cancer cause-specific mortality. The cancer diagnosis and outcome was verified by reviewing clinical records and death certificates. Adherence to aMED was assessed by the use of the dietary information collected in two-yearly intervals with a FFQ. The aMED score, adapted from the traditional MD Trichopolou score, awards one point for intakes higher than the population median of vegetables, legumes, fruits, nuts, whole grains, fish, and monounsaturated:saturated fat ratio, and intakes less than the median in meat, and in alcohol if intake is outside a given range [64]. No statistically significant associations were found between high adherence to aMED (vs. low adherence) and breast or colorectal cancer mortality in this study, nor did the authors observe any association with regard to all-cause mortality in the breast or colorectal cancer survivors. All estimates were adjusted for relevant prognostic factors of the disease (e.g., tumor stage, cancer site, and treatment), as well as for the patient´s lifestyle or dietary habits before the cancer diagnosis. Another American cohort study, conducted within the HPFS, addressed the association between aMED and cancer mortality, particularly concerning prostate cancer mortality [58]. In this study, 4538 early-stage prostate cancer patients were followed-up since the cancer diagnosis until mortality or end of follow-up. Comparable methods were applied to collect dietary information and to ascertain cancer diagnosis and death. Likewise, by adjusting for prognostic factors of the disease and lifestyle/dietary habits before the diagnosis, a significant inverse association was observed between high vs. low adherence to the MD (both as aMED and MDS) and mortality from any cause (HR = 0.78; 95% CI: 0.67–0.90). However, statistical significance was not reached for mortality from prostate cancer.

Other post-diagnosis dietary indices that have been examined in relation to MD were the modified Mediterranean diet score (mMDS) and the traditional Mediterranean diet score (MDS), which are similar in their composition with respect to aMED [64]. The German PopGen Biobank Study [56], which included 1404 colorectal cancer patients (histologically confirmed), among which 204 died during six years of follow-up, reported that high vs. low adherence to the mMDS was associated with a reduced all-cause mortality risk among those patients. The study accounted for the influence of all possible prognostic factors and of pre-diagnostic adherence to the MD. Adherence to the MD by the mMDS at pre- and post-diagnosis was assessed using dietary information collected via FFQs administered in both time intervals. Lastly, within the National Health and Nutrition Examination Survey (NHANES) study [59], 240 participants who self-reported a cancer diagnosis of breast or gynecological cancers were followed-up during 10 years on average. Cancer mortality was assessed by death certificates from clinical records and mortality registers. Dietary information collected with a single 24 hour recall (HR) was used to score the participants into levels of adherence to the MDS. This study showed that high vs. low adherence to this score was not significantly associated with neither all-cause mortality nor breast cancer mortality.

It is worth noting that four of these studies also accounted for pre-diagnosis dietary information on PBDPs [55,58,59,63], which was considered for the adjustment of baseline dietary intake in some of these studies [58,63]. Only the study by Kim et al. [55] examined the effect of pre-diagnosis PBDPs (aMED score) on cancer mortality among cancer survivors. As a result, no association was reported for total mortality, breast cancer mortality, distant recurrences or non-breast cancer mortality.

Results of the meta-analyses of these studies by cancer site were only possible for breast and colorectal cancer. No study evaluated the association between post-diagnosis PBDPs and cancer mortality in relation to all-cause mortality. Publication bias was unlikely according to funnel plots and Egger test.

#### 3.4.1. Breast Cancer Mortality

Of the three studies evaluating an association between a PBDP and cancer mortality, there were two cohort studies considering an MD score and breast cancer mortality [55,59]. The intervention study was not considered for the meta-analysis due to its different nature. Pooled estimates for the 2849 breast cancer patients revealed that adherence to the MD (high vs. low) was not associated with all-cause mortality among breast cancer survivors (RR = 0.87; 95% CI: 0.85–1.01). While heterogeneity between the studies was not apparent (I^2^ = 0%, *p* = 1), the study by Karavisouglou contributed with a small number of breast cancer patients, whose diagnoses were not confirmed [59]. Moreover, in this study, breast and gynecological cancers were considered together, thus biasing the pooled analyses to some extent. The other two studies provided more robust results [55,57], and supported that a PBDP from breast cancer onset reduces all-cause or breast cancer mortality.

#### 3.4.2. Colorectal Cancer Mortality

The studies by Fung [54] and Ratjen [56] were the only two studies available on the association between PBDP and cancer mortality in colorectal cancer survivors. The MD was considered as a plant-based pattern in both studies, either as aMED or mMDS. Together, these studies combined the results of 2605 colorectal cancer patients. The summary estimates for all-cause mortality revealed a nonsignificant association (HR = 0.66; 95% CI: 0.37–1.17); no other outcome events could be analyzed due to lack of data. However, there was heterogeneity between both studies (I^2^ = 79%, *p* = 0.03), possibly driven by differences inherent to the study populations (country of recruitment, gender distribution, and clinical characteristics of the patients). It was not possible to examine the influence of these variables on the pooled results.

## 4. Discussion

The present review is the first to address the available evidence on the association between PBDPs and cancer-related outcomes, including overall cancer mortality, cause-specific mortality, and cancer recurrence. Plant-based diets have been traditionally regarded as vegetarian diets, but other definitions of plant-based foods can be considered by rating negatively some or all animal foods, and even accounting for the quality of plant-based foods in the diet. Only adherence of the MD was found to reduce overall cancer mortality risk, but none of the studies accounted for the influence of prognostic factors on this association; thus, the true independent effect of the MD with cancer mortality risk remains inconclusive. Studies on other PBDPs in relation to these cancer outcomes have provided nonsignificant or ambiguous results. The same was found for post-diagnostic studies on these associations. Therefore, for the considered cancer-related outcomes, there is still insufficient evidence for asserting that PBDPs help in reducing the risk of these outcomes.

Plant-based diets are dietary sources of several bioactive compounds such as fiber, phenol, polyphenol, and sulfur compounds, and other antioxidants compounds including vitamins [7]. In the literature, foods of vegetable origin have been associated with cancer mortality outcomes, although conflicting results have been reported. These include, for example, legumes [9], fruits and vegetables [15], nuts [16], whole grains [16], and olive oil [18]. Bioactive compounds in these foods, however, have been shown to have anti-carcinogenic effects in experimental models and epidemiological studies [7]. For instance, high intake of fiber and polyphenols from cereals have been shown to reduce cancer mortality risk [65,66,67]. As for cancer incidence, a two-tier mechanism could explain how these compounds could prevent cancer mortality risk: first, fiber, mostly soluble fiber, modulates the gut microbiota composition improving the colonic barrier functions, and second, substrates such as resistant starch, non-starch polysaccharides (β-glucans), and phenols are metabolized into active metabolites by the commensal microbiota [68]. The resulting metabolites have well known anti-cancer effects and could likewise prevent cancer mortality. For instance, phenolic acids are antioxidants with free-radical scavenging activity. Free radicals play an important role not only for cancer development but also in cancer treatment since anti-cancer drugs generate reactive oxygen species themselves [69]. Therefore, neutralizing their damaging effects is crucial to reduce mortality and secondary outcomes in cancer patients. Other metabolites, such as short-chain fatty acids are also major players in the maintenance of gut integrity and immune homeostasis, to prevent tumor development [68].

The drawbacks of individual nutrient or foods studies have been repeatedly described [19]. PBDPs or indexes based on the potential beneficial effects of various dietary factors, and accounting for the interaction of all phytochemicals contained in the diet, should be therefore a better approach to examine the association between plant-based diets and cancer mortality. PBDPs have been related to a low-risk immunological profile (lower C-reactive protein, fibrinogen, and leukocyte levels) [70], lower adiposity markers [62], and better antioxidant status [71]. These are probably the underlying mechanisms by which these diets could improve the immune response in cancer patients, while also slowing tumor growth and risk of developing subsequent events.

Vegetarian diets are based on the consumption of plant-based foods, namely vegetables, fruits, whole grains, legumes, nuts, and seeds, with the elimination of foods of animal origin such as meat, poultry, wild game, seafood, and their byproducts [72]. In a narrower sense, vegetarian diets may or may not include eggs and dairy products. Vegan diets, for instance, exclude these dietary factors, whereas lacto-, ovo-, and lacto-ovo vegetarians include either dairy products, or eggs and egg products, or both dairy products and eggs, respectively. Vegetarian diets comply with healthy diet recommendations and are therefore considered a healthful dietary pattern [72]. A negative side of this dietary pattern is the potential deficient intake of certain nutrients such as iron and vitamin B12. Long-chain n-3 fatty acids, eicosapentaenoic acid (EPA), and docosahexaenoic acid (DHA), are lower in vegetarians and typically absent in vegans [73]. Decreased intakes of some of these nutrients have been related to a higher cancer incidence in some studies [74]. On the contrary, the reduced intake of heme-iron in vegetarians is presumed to be beneficial taking into consideration that high ferritin levels due to iron overload have been related to a higher risk of certain cancers [75]. However, it is considered that if vegetarian diets are well-planned, their naturally high content of beneficial bioactive compounds outweigh the negative sides [72,76]. Indeed, it has been reported that vegetarians have a lower cancer incidence than nonvegetarians [26], or at least regarding colorectal cancer for semi-vegetarians or pesco-vegetarians compared to nonvegetarians [77]. While the results of our review tended to suggest that vegetarians and vegans have a lower risk of cancer mortality and cancer-specific mortality when compared to nonvegetarians, we did not observe significant associations in meta-analyses combining results of the selected studies. By updating a previous review examining vegetarian diets regarding several health issues [26], we provide more consistent data, although it is still insufficient to provide meaningful conclusions. Indeed, vegetarians do not make up large population groups and are therefore difficult to ascertain in epidemiological studies. Moreover, as shown in the study by Key et al. [60], vegetarians or vegans often adopt a healthier lifestyle (e.g., non-smoking) as compared to omnivores, making it more difficult to establish comparisons between both groups. The sample size of the included studies was too limited to permit overall and cause-specific cancer mortality studies. Another important aspect as to why we probably did not observe a significant overall effect might be due to different considerations of vegetarians, vegans, and type of vegetarian diets among the studies. Based on the responses to a finite list of foods, participants were classified into vegetarians and vegans in most studies; only the study by Orlich based this classification upon a validated FFQ [61]. Misclassification of vegetarian diets was therefore likely in these studies. In addition, it was not possible to make any comparisons with regard to type of vegetarian diet since few studies attempted to link the type of diet with cancer mortality risk [38]. Thus, for example, whether pesco-vegetarian or lacto-vegetarian diets, rich in dietary factors with colorectal cancer prevention potential (e.g., calcium, fiber, n-3 fatty acids, and vitamin antioxidants), have a stronger prevention effect against mortality of this disease could not be confirmed.

Recently, a priori-defined plant-based scores have been proposed to allow comparisons between different levels of compliance with vegetable-derived foods against animal-derived foods [53]. Provegetarian food pattern scores are newly developed tools to assess the level of adherence to a vegetarian-like diet (i.e., preference of plant-based foods). Unlike the vegetarian dietary patterns, these scores consider moderate intakes of animal foods such as fish, poultry, and dairy in the assessment of the score, under the assumption that moderate intake of these foods may confer some health benefits [78,79,80]. Moreover, in this manner, these scores are presumed to overcome the potential nutrient intake deficiencies (e.g., vitamin B12), that a strict plant-based diet such as the vegetarian diet poses. Accordingly, they score vegetable-derived foods positively, while all animal-derived foods are scored negatively. However, with only two studies evaluating how these scores affect cancer mortality risk [52,53], it has been difficult to arrive at any concrete conclusion. The time point considered for the assessment of the provegetarian diet differed in both studies. Furthermore, both studies were not comparable in the sense that the provegetarian diet score was not defined in the same way. While both considered all animal foods, only the study by Baden et al. [53] distinguished the quality of plant-based foods in the scoring. This study reported associations for a standard provegetarian diet score, and for its healthy and unhealthy versions. Relative to the unhealthy score, the healthy one scored more negatively foods rich in refined carbohydrates, which are supposed to be implicated in cancer risk through energy metabolism, insulin, and insulin-like growth factor (IGF-1) upregulations. A major impact of the healthy provegetarian diet score for cancer mortality prevention was therefore expected. Nevertheless, significant associations were only observed for the standard score.

The MD encompasses the traditional dietary pattern found in the olive-growing regions of the Mediterranean basin in the 1960s [81], and is globally recognized as a healthy dietary model [82]. The traditional MD pattern is a well-defined plant-based dietary pattern characterized by: the daily use of olive oil; an abundance of plant foods such as fruits and vegetables, nuts and seeds, cereals and legumes; the consumption of fish and seafood especially in coastal regions; moderate-to-low intake of dairy products mostly from fresh cheese and yogurt; moderate alcohol mostly in the form of wine; and a less frequent consumption of meat and meat products [83]. Thus, the MD pattern is distinctively plant-based, and thus a valuable alternative to the vegetarian diet, as it provides a good supply of fiber, phytochemicals, vitamins, and minerals, even closing some nutritional gaps of the vegetarian diet [83]. For the reasons cited above, we considered the MD as a reference PBDP. Moreover, the MD is presumed to boost the endogenous antioxidant defense and the immune system to prevent cancer and, possibly, also fatal outcomes of this disease. With respect to cancer incidence, indeed, as is also the case with other dietary patterns, the MD is an established dietary pattern for cancer prevention [13,84]. Other dietary patterns resembling plant-based diets such as the very low ketogenic diet, seem to have cancer prevention potential through weight loss and related mechanisms, but have been scarcely examined with regard to cancer mortality [85].

Since the first definition of an MD index, multiple adaptations have been created to improve its conformity to the MD [64]. The MD scores used to assess the association between the MD and cancer mortality risk are the MDS, MDP, rMED, and aMED. Pooled analyses of the included studies revealed a significant inverse association between high vs. low adherence to the MD and overall cancer mortality risk. While we combined the studies regardless of the MD score, it has been reported that there is only moderate concordance between them [86]. Indeed, MD scores vary largely in the types of foods included and the intake levels in the population. Thus, variability in the scoring schemes of the MD scores might have affected our results. However, when restricting our meta-analysis to studies using the MDS (the predominant score), the result was maintained (data not shown). Despite the fact that our results support a beneficial effect of the MD for the prevention of cancer mortality, we have to be cautious in the interpretation of these results. No study considered the influence of relevant factors related with the disease outcomes, such as treatment, on the associations. In addition, there was scarce data on the association between MD and cancer mortality by type of cancer.

As outlined before, most of the studies on the association between PBDPs and cancer mortality considered the influence of such diets from pre-diagnosis states to the event outcomes. However, post-diagnosis diet might have a strong short-term or even long-term effect on cancer mortality and other related outcomes. Whether eating a PBDP after cancer diagnosis affects cancer outcomes has been addressed in several studies. Although these studies accounted for a large number of factors related to the outcome (all potential prognostic factors: clinical and pathological tumor characteristics, treatment, and comorbidities), their results must be interpreted with caution due to the limited sample sizes. Moreover, there was only one intervention study on the effect of a well-defined plant-based diet on cancer-related outcomes [57]. Importantly, this study did not address whether consuming the high vegetable/fruit/fiber and low-fat diet of the study intervention early in life would alter the risk of breast cancer and cancer mortality as well. The remaining studies were cohort studies assessing the association between adherence to the MD since the cancer diagnosis and cancer mortality [34,35,36,37,38,39,40,41,42,43,44,45,46,47,48,49,50,51,52,53,54,55,56,58,59]. Some [34,35,36,37,38,39,40,41,42,43,44,45,46,47,48,49,50,51,52,53], but not all [54,55,56,57,58,59], of the included studies also accounted for dietary behaviors before the cancer onset. In fact, as this review shows, dietary patterns over a lifetime are likely to influence cancer mortality. Future studies on the association between PBDPs and cancer mortality should therefore consider assessing adherence to PBDPs throughout the lifespan, to cover pre- and post-diagnostic influences of these dietary patterns on the associations.

Thus, while dietary patterns have been consistently associated with a reduction of cancer incidence, an association with cancer mortality has not been clearly established. This contradiction could be explained by the fact that cancer mortality is greatly influenced by the type and treatment regime. Indeed, cohort studies assessing the association from a pre-diagnosis state typically did not account for this factor. Rather, this was accounted for in studies evaluating how adherence to these plant-based diets since diagnosis impacts on these outcomes. On the other hand, the lack of statistical power to detect significant associations in cause-specific mortality studies is another important limiting factor that makes it impossible to draw any meaningful conclusions on this association. In fact, there are no studies on cause-specific cancer mortality other than lung cancer, breast cancer, colon cancer, and prostate cancer. Also of note, is the fact that studies on other cancer-related outcomes, such as secondary cancers, are unavailable. No study considered the interaction between dietary and genetic factors on these associations. Most studies relied on a single measurement of diet at baseline, but dietary patterns may change over time and the length of exposure to plant-based-like dietary patterns may account for the differences observed between results from different cohorts. Dietary assessment tools (3-day records, 24-h recalls or FFQs) also differed greatly between the studies, as well as the studied populations. For instance, there were both pre and postmenopausal breast cancer patients considered jointly in some studies. Moreover, while the definitions of vegetarians and vegans were well-defined, some studies seemed to mix up vegans and vegetarians. Further, in the case of the comparison groups, there were differences among the studies with regard to the definition of meat eaters (omnivores). The included studies considered either populations of adherers to a specific PBDP (for example, vegans and vegetarians) or cohorts with individuals classified into different levels of adherence to PBDPs (for example, MD adherence groups) according to their reported dietary data. Therefore, the possibility that some studies misclassified vegetarians, vegans or adherence to other types of plant-based diets cannot be ruled out. Finally, plant-based diets contain a huge variability of bioactive compounds depending on the dietary source, which makes the definition of a healthy plant-based diet particularly complex. Among the limitations of this review, there were also issues related to population overlap in the vegetarian studies [35,38], which could have introduced bias to our results. We could also not analyze whether duration of adherence to any PBDP had any influence on the results.

There are also strengths of this review worthy of consideration: this is the first review and meta-analysis that has analyzed all the available data on PBDPs in relation to cancer mortality and related outcomes. While a rapid review was conducted, we complemented the identification of studies with those included in other reviews related to our topic. However, a quality assessment of these studies was not performed. We were able to conduct analyses by some cancer types and to evaluate the influence of some study characteristics on the results. However, we were not able to meta-analyze studies according to all cancer types due to lack of studies.

## 5. Conclusions

This review and meta-analysis of the current available evidence on the association between PBDPs and cancer mortality show that there is limited evidence regarding the beneficial effects of vegetarian diets for the prevention of cancer-related outcomes in the general population and in cancer survivors. This review also shows that there is suggestive evidence regarding the association between the MD pattern and cancer mortality. There were very few studies evaluating how these dietary patterns influence cancer mortality after the cancer diagnosis. Breast cancer and colorectal cancer outcomes were by far the most studied cancer types, but the number of studies is small. Thus, there is an urgent need to increase our knowledge on the usefulness of plant-based diets for the prevention of cancer mortality. Well-designed studies, considering consensus definitions of PBDPs and all pertinent factors including prognostic factors of the disease, genomics, and others, are needed to determine the effect of plant-based diets on cancer survival and cancer recurrence, before and after the diagnosis of cancer.

## Figures and Tables

**Figure 1 nutrients-12-02010-f001:**
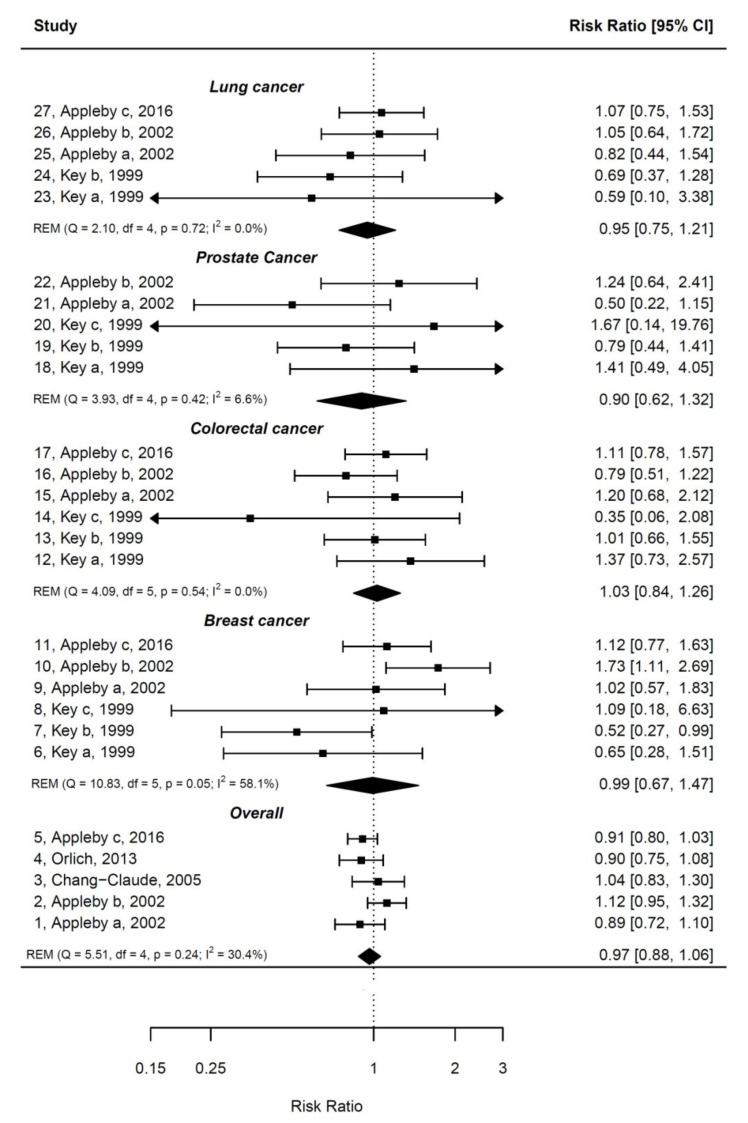
Meta-analysis under random-effects model (REM) with regard to vegetarian diet (VD) (vs. nonvegetarian diet) by subgroups of overall cancer mortality and cause-specific mortality. Q and I^2^ statistics are indicated for each subgroup analysis together with the pooled estimate (rhombus).

**Figure 2 nutrients-12-02010-f002:**
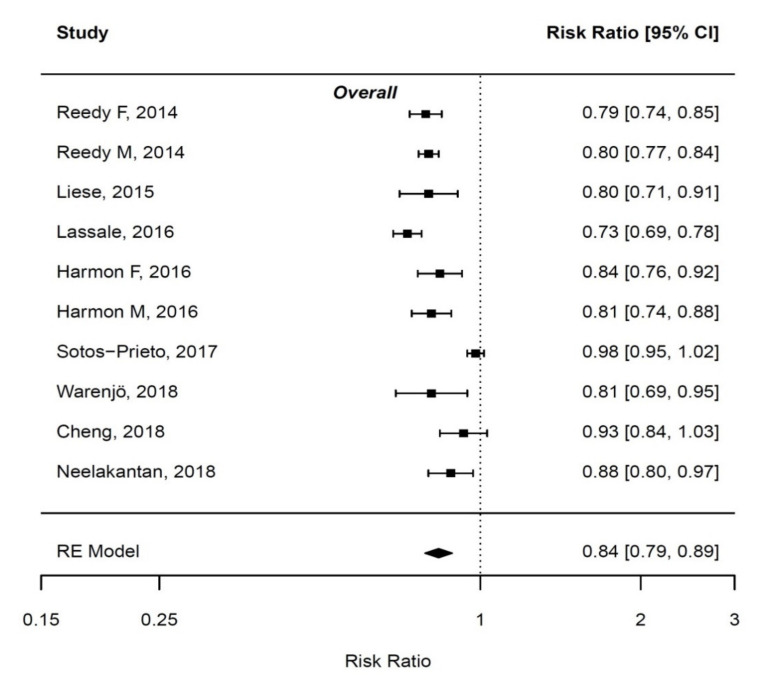
Meta-analysis under random-effects model (REM) with regard to adherers to the Mediterranean diet (MD) (high vs. low) and overall cancer mortality. F = females, M = males.

**Table 1 nutrients-12-02010-t001:** Characteristics of the studies evaluating VD (vs. nonvegetarian diet) and cancer mortality including cause-specific cancer mortality.

N	Author (Year)	Country	Study (Design)	Age, Year Sex	FU, Year	*n/N*	Outcome	RR (95% CI)	Adjustment	Dietary Assessment	Vegans and Vegetarians Ascertainment
**1**	Key et al. (1999) [34]	USA	Adventist Mortality Study (Cohort)	52.5 M/F	5.6	41/10,258	CRC	1.37 (0.73–2.56)	1, 2, 3	NA	vegetarians: those who reported that they did not eat any meat or fish; non-vegetarians: all others
						6/10,258	Lung	0.59 (0.10–3.28)			
						26/10,258	Breast	0.65 (0.28–1.52)			
						15/10,258	Prostate	1.41 (0.49–4.04)			
	Key et al. (1999) [34]	USA	Adventist Health Study-1 (Cohort)	52.5 M/F	11.1	104/8003	CRC	1.01 (0.66–1.56)	1, 2, 3	NA	
						96/8003	Lung	0.69 (0.37–1.27)			
						64/8003	Breast	0.52 (0.27–0.97)			
						66/8003	Prostate	0.79 (0.44–1.41)			
	Key et al. (1999) [34]	Germany	Heidelberg Study (Cohort)	46.5 M/F	9.9	5/1083	CRC	0.35 (0.06–2.11)	1, 2, 3	NA	
						5/1083	Breast	1.09 (0.18–6.67)			
						3/1083	Prostate	1.67 (0.14–19.6)			
**2**	Appleby et al. (2002) [35]	UK	Oxford Vegetarian Study (Cohort)	42.3 M/F	17.6	156/4674	Overall	0.89 (0.72–1.10)	1, 2, 3	NA	vegetarians: those who reported that they did not eat any meat or fish; non-vegetarians: all others
						25/4674	CRC	1.20 (0.68–2.13)			
						16/4674	Lung	0.82 (0.44–1.56)			
						22/4674	Breast	1.02 (0.57–1.84)			
						8/4674	Prostate	0.50 (0.22–1.17)			
	Appleby et al. (2002) [35]	UK	Health Food Shopper Study (Cohort)	42.8 M/F	18.7	203/4600	Overall	1.12 (0.95–1.32)	1, 2, 3	NA	vegetarians: those who answered yes to the question “Are you a vegetarian?”
						33/4600	CRC	0.79 (0.51–1.22)			
						24/4600	Lung	1.05 (0.64–1.72)			
						41/4600	Breast	1.73 (1.11–2.69)			
						16/4600	Prostate	1.24 (0.64–2.41)			
**3**	Chang- Claude et al. (2005) [36]	Germany	Heidelberg Study (Cohort)	50 M/F	21	76/1225	Overall	1.04 (0.86–1.34)	1, 2, 3, 4, 5, 6, 7	Questions on the frequency of consumption of vegetables, fruits, nuts, cereal, milk products, eggs, fish, meat, and processed meat	vegan: avoid meat, fish, eggs, and dairy products; lacto-ovo vegetarian: avoid meat and fish but eat eggs and/or dairy products; nonvegetarian: occasionally or regularly eat meat and/or fish
**4**	Orlich et al. (2013) [37]	USA	Adventist Health Study-2 (Cohort)	57.5 M/F	5.9	706/73,308	Overall	0.90 (0.75–1.09)	2, 3, 4, 5, 6, 8, 9, 10, 11, 12	Self-administered quantitative FFQ of more than 200 food items	vegan: eggs/dairy, fish, and meats <1 month; lacto-ovo vegetarian: eggs/dairy ≥1 month but fish and meats <1 month; pesco-vegetarian: fish ≥1 month but meats <1 month; semi-vegetarians: non-fish meats ≥1 month and meats combined ≥1 month but <1 time/wk; nonvegetarians: all others
**6**	Appleby et al. (2016) [38]	UK	Oxford Vegetarian Study (*N* = 52,659); European Prospective Investigation into Cancer and Nutrition (EPIC) Oxford Vegetarian Study (*N* = 2708) (Cohorts)	44.8 M, 43.0 F	>5 years	520/55,367	Overall	0.91 (0.80–1.03)	1, 2, 3, 4, 5, 7, 10, 22, 23, 24, 25, 26, 27, 28	Questions on whether or not they consumed meat, fish, dairy products, and eggs, and two questions on the frequency of meat consumption	four diet groups: regular meat eaters (eat meat on >5 times/w), low meat eaters (eat meat <5 times/w), fish eaters, and vegetarians and vegans combined (non-meat and/or fish eaters); nonvegetarians: all others
						76/55,367	CRC	1.11 (0.79–1.58)			
						20/55,367	Pancreas	0.44 (0.26–0.76)			
						62/55,367	Lung	1.07 (0.75–1.54)			
						70/55,367	Breast	1.12 (0.77–1.63)			
						41/55,367	Ovary	0.97 (0.61–1.52)			
						28/55,367	Hematolymphoid	0.47 (0.30–0.73)			

RR = relative ratio; CI = confidence interval; F = females; M = males; FFQ = food frequency questionnaire; 24-HR = 24 h recall; CRC = colorectal cancer; NA = not available; n = number of events; *N* = total population (or vegetarians). Adjustments: 1: sex; 2: sge; 3: smoking; 4: physical activity; 5: alcohol intake; 6: education; 7: body mass index (BMI); 8: race; 9: income; 10: marital status; 11: region; 12: sleep; 13: hormone replacement therapy (HRT) use; 14: history of peptic ulcer and inflammatory bowel disease (IBD); 15: family history of colorectal cancer; 16: energy; 17: treatment for diabetes; 18: aspirin/statins; 19: supplemental Ca/vitamin D; 20: fiber; 21: colonoscopy; 22: supplements; 23: study/method of recruitment; 24: parity; 25: oral contraceptive use; 26: HRT use; 27: diabetes; 28: blood pressure.

**Table 2 nutrients-12-02010-t002:** Characteristics of the studies evaluating provegetarian diets and cancer mortality including cause-specific cancer mortality.

Author (Year)	Country	Study (Design)	Age, Year Sex	FU, Year	*n/N*	HR/RR (95% CI)	Adjustment	Diet Assessment	Provegetarian Diet Assessment
Martinez- González et al. (2014) [52]	Spain	(Prevención con Dieta Mediterránea (PREDIMED) (Cohort)	63 M/F	4.8	130/7216	High vs. low adherence Overall 0.66 (0.35–1.24)	1, 2, 3, 4, 5, 6, 9	FFQ of 137 items (administered every year)	Points assigned by quintiles (1 to 5) of consumption, whereby fruits, vegetables, nuts, cereals, legumes, olive oil, and potatoes are positive components, and animal fats, eggs, fish, dairy products, and meats or meat products are negative components (scoring reversed)
Baden et al. (2019) [62]	USA	Nurses’ Health Study (NHS); Health Professionals Follow-up Study (HPFS) (Cohorts)	67 M/F	12	4263/1,096,638; 1140/1,096,638; 374/1,096,638; 262/725,316; 175/371,322	Per 10 points increase in adherence Overall 0.93 (0.89–0.98) Lung 0.97 (0.85–1.10) NHS 0.77 (0.62–0.94) HPFS Colon 1.15 (0.90–1.47) NHS 0.77 (0.57–1.03) HFPS Breast 1.11 (0.89–1.39) NHS Prostate 0.73 (0.55, 0.96) HFPS	1, 2, 3, 4, 5, 7, 8, 9, 10, 11, 12, 13, 14, 15, 16	Semi-quantitative FFQ (administered every 4 years)	Points assigned by quintiles (1 to 5) of consumption with regard to all plant-based foods (standard score), or either healthy plant-based foods (healthy score) or unhealthy plant-based foods (unhealthy score). Scoring is reversed for all other animal-based foods.

F = females; M = males; FFQ = food frequency questionnaire; 24-HR = 24 h recall; CRC = colorectal cancer; NA = not available; *n* = number of events; *N* = total population. Adjustments: 1: sex; 2: age; 3: smoking; 4: physical activity; 5: alcohol intake; 6: education; 7: BMI; 8: race; 9: energy intake; 10: use of aspirin and multivitamins; 11: family history of medical conditions; 12: menopausal status; 13: HRT use; 14: weight change; 15: medical history; 16: use of medication for hypertension and hypercholesterolemia.

**Table 3 nutrients-12-02010-t003:** Characteristics of the studies evaluating the Mediterranean diet (high vs. low adherence) and cancer mortality including cause-specific cancer mortality.

Author (Year)	Country	Study (Design)	Age, Year Sex	FU, Year	*n/N*	Outcome	RR/HR (95% CI)	Adjustment	Dietary Assessment	Ascertainment (a Priori MD Index)
Lassale et al. (2016) [39]	Europe	European Prospective Study into Cancer and Nutrition (EPIC) (Cohort)	50.8 ± 9.8 F/M	12.8	451,256/7475	Overall	High vs. low adherence MDS: 0.75 (0.70–0.80) rMED: 0.73 (0.69–0.78)	1, 2, 3, 4, 5, 6	Usual diet over the previous 12 months assessed at baseline using validated dietary questionnaires	MD score (MDS) and relative MD score (rMED)
Vargas et al (2016) [40]	California and Hawaii	Multiethnic Cohort Study (MEC) (Cohort)	F: 69–74.5 M: 68–75 F/M	6	>200,000 4204 CRC Deaths: 1976 all-cause 1095 CRC	CRC	High vs. low adherence 0.90 (0.57–1.43) Per 1-SD increase in adherence F: 0.74 (0.54–1.01) M: 1.07 (0.81–1.42)	1, 2, 3, 5, 7, 8, 9, 10, 11, 12, 13, 14	self-administered baseline quantitative FFQ (180 items)	alternative MD index (aMED)
Harmon et al. (2015) [44]	California and Hawaii	Multiethnic Cohort Study (MEC)(Cohort)	45–75 F/M	13–18	F: 145612 M: 70170 Cases: F: 5030 M: 5853	Overall	High vs. low adherence F: 0.84 (0.76–0.92) M: 0.81 (0.75–0.89)	1, 2, 3, 4, 5, 8, 15, 16, 17, 18	self-administered baseline quantitative FFQ (180 items)	alternate MD (aMED)
Liese et al. (2015) [45]	USA	National Institutes of Health—American Association of Retired Persons (NIH-AARP) Diet and Health Study, the MEC study, and Women Health Initiative Observational Study (WHI-OS) (Cohorts)	Early 60s F/M	≥10	645,272/ >5000	Overall	High vs. low adherence WHI-OS: F 0.80 (0.70–0.90) NHI-AARP: 0.80 (0.75–0.85) MEC: 0.85 (0.75–0.90)	1, 2, 3, 4, 5, 7, 8, 15, 16, 17, 18, 19	self-administered FFQ that assessed dietary intake over the past year or past 3 months (WHI-OS)	alternate MD (aMED)
Cuenca-García et al. (2014) [46]	Spain	Aerobics Center Longitudinal Study (ACLS) (Cohort)	20–84 F/M	11.6	12,449/134 F/M	Overall	High vs. low adherence 1.63 (0.91–2.92)	1, 2, 3, 8, 16, 20, 21, 22	3-day diet record	MD score (MDS)
Reedy et al. (2014) [47]	US	NIH-AARP Diet and Health Study (Cohort)	50–71 F/M	15	F: 182,342 M: 242,321 Cases:F: 10,769 M: 18,646	Overall	High vs. low adherence F: 0.79 (0.74–0.85) M: 0.80 (0.77–0.84)	1, 2, 3, 4, 5, 7, 8, 15, 16, 17, 18, 19	124-item validated FFQ	alternate MD (aMED)
Cheng et al. (2018) [48]	USA	Iowa Women’s Health Study (IWHS) (Cohort)	55–69 F	26	41,836/4665	Overall	High vs. low adherence 0.93 (0.84–1.03)	1, 2, 3, 4, 5, 8, 15, 18, 33, 35, 36	127-item semi-quantitative validated FFQ	MD pattern (MDP)
Neelakan-tan et al. (2018) [49]	China	Singapore Chinese Heath Study (SCHS) (Cohort)	45–74 F/M	17	63,257/5306	Overall	High vs. low adherence 0.88 (0.80–0.97)	1, 2, 3, 4, 5, 8, 17, 20, 37, 38	165-item validated FFQ	alternate MD (aMED)
Warensjö et al. (2018) [50]	Sweden	Swedish Mammography Cohort (Cohort)	61 F	17	33,341/2355	Overall	High vs. low adherence 0.81 (0.69–0.94)	2, 3, 5, 8, 35, 39, 40	96-item validated FFQ	modified MD score (mMDS)
Sotos-Prieto et al. (2017) [51]	US	Health Professionals Follow-up Study (HPFS) and NHS (Cohort)	30–75 F/M	16	F: 47,994/2089 M: 25,745/1226	Overall	High vs. low adherence 0.98 (0.93–1.0)	1, 2, 3, 4, 18, 19, 23, 28, 35, 36, 41, 42, 43, 44, 45, 46	130-item validated FFQ	alternate MD (aMED)
Vormund et al. (2015) [41]	Switzerland	National Research Program (Cohort)	16–92 F/M	21.4	17,861/1347	Overall	Per 1 point increase in adherence 0.96 (0.94–0.98)	1, 2, 4, 15, 20, 48, 49, 50	one 24-HR	MD score (MDS)
Buckland et al. (2011) [42]	Spain	EPIC (Cohort)	29–69 F/M	13.4	40,622/1855	Overall	High vs. low adherence 0.92 (0.75–1.12)	1, 2, 3, 4, 5, 8, 20, 49, 53	diet history questionnaire	relative MD score (rMED)
Lagiou et al. (2006) [43]	Sweden	Uppsala Health Care Region (Cohort)	30–49 F	12	42,237/572	Overall	Per 1 point increase in adherence <40 years: 1.07 (0.79–1.43) ≥40 years: 0.84 (0.71–1.01)	1, 2, 3, 4, 5, 8, 54, 55, 56, 57, 58	80-item validated FFQ (6 months before)	MD score (MDS)

HR = hazard ratio; SD = standard deviation; F = females; M = males; FFQ = food frequency questionnaire; 24-HR = 24 h recall; CRC = colorectal cancer; NA = not available; n = number of events; *N* = total population. Adjustments: 1: age; 2: smoking; 3: physical activity; 4: BMI; 5: education; 6: dietary score at baseline; 7: ethnicity; 8: total energy intake; 9: radiation treatment; 10: pack-years; 11: chemotherapy; 12: nonsteroidal anti-inflammatory drug (NSAID) use; 13: family history of colorectal cancer; 14: comorbidities; 15: marital status; 16: alcohol intake, 17: type 2 diabetes; 18: postmenopausal hormone replacement therapy; 19: race, 20: sex; 21: baseline examination year; 22: abnormal electrocardiogram; 23: dietary supplement use; 24: oral contraceptive use; 25: stage; 26: time since diagnosis; 27: age at first birth and parity; 28: menopausal status; 29: eeight change; 30: categories of treatment; 31: time between diagnosis and completion of the questionnaire; 32: socioeconomic status; 33: self-reported prevalent chronic diseases at baseline; 34: moderate to vigorous physical activity; 35: diet score; 36: family history of cancer in a first-degree relative; 37: sleep duration; 38: history of hypertension; 39: living alone; 40: Charlson comorbidity index; 41: aspirin use, 42: changes in smoking status; 43: family history of myocardial infarction; 44: family history of diabetes; 45: changes of physical activity; 46: changes in total energy intake; 47: weight change; 48: survey wave; 49: region; 50: nationality; 53: waist circumference; 54: potato intake; 55: egg intake, 56: polyunsaturated lipid intake; 57: sweet intake; 58: non-alcoholic beverage intake.

**Table 4 nutrients-12-02010-t004:** Characteristics of the studies evaluating plant-based dietary patterns at post-diagnosis and cancer mortality and recurrence.

Author (Year)	Study (Country)	Patients Sex/N	Study Design Follow-Up	Dietary Assessment	Outcome and *N* Events	HR (95% CI)	Adjustment
Pierce, et al. (2007) [57]	Women´s Healthy Eating and Living (WHEL) Study (USA)	F (18–70 years)/1537 F in the intervention group and 1551 F in the control group, all having been treated for early-stage breast cancer (stage I–IIIA) according to clinical records	Intervention study (randomized) FU = 7.3 years	Intervention based on advising a diet high in vegetables, fruits, and fiber, and low in total fat (5 vegetable servings, 16 oz of vegetable juice, 3 fruit servings, 30 g of fiber, and 15% to 20% of energy intake from fat); control group received “5-A-Day” dietary guidelines; Four 24-HRs during the intervention	(1) Mortality from any cause: 155 deaths in the intervention group vs. 160 deaths in the control group (2) Recurrence (invasive): 256 events in the intervention group vs. 262 events in the control group	0.91 (0.72–1.15) Disease-free survival 0.96 (0.80–1.14)	1, 2, 3, 4, 6, and diet at baseline in sensitivity analyses
Kim et al. (2011) [55]	Nurses’ Health Study (NHS) (USA)	F/2729 (30–55 years), diagnosed with breast cancer (stage I–III) during 1978–1998 according to clinical records, alive and without recurrences for at least 1 year	Cohort study FU = 6–24 years	130-item FFQ (administered every 2 years, at least 12 months after cancer diagnosis) aMED (alternate MD)	High vs. low adherence aMED (1) Mortality from any cause (*N* = 572): 123 vs. 117 deaths 2) Mortality from breast cancer (*N* = 302): 74 vs. 51 deaths (3) Mortality from non-breast cancer (*N* = 270): 49 *vs.* 66 deaths	0.87 (0.64–1.17) 1.15 (0.74–1.77) 0.80 (0.50–1.26)	3, 6, 7, 9, 10, 11, 12, 13, 14, 15, 16
Fung et al. (2014) [54]	NHS (USA)	F/1201, diagnosed with CRC (stage I–III) during 1986–2008 according to clinical records, and alive for at least 6 months	Cohort study FU = 11.2 years	Semi-quantitative 130-item FFQ (administered every 2 years; at least 6 months after cancer diagnosis) aMED	High vs. low adherence aMED (1) Mortality from any cause (*N* = 435): 88 vs. 113 deaths (2) Mortality from CRC (*N* = 162): 36 vs. 39 deaths	0.87 (0.63–1.21) 0.84 (0.50–1.42)	3, 4, 6, 7, 10, 13, 17, 18, 19, 21, 22, and baseline diet
Ratjen et al. (2017) [56]	German PopGen Biobank Study (Germany)	F,M/1404 (mean age = 62 years), diagnosed with invasive CRC during 1998–2005 according to clinical records	Cohort study FU = 6 years	112-item semi-quantitative FFQ modified Mediterranean diet score (mMDS)	High vs. low adherence mMDS (1) Mortality from any cause (*N* = 204): 34 vs. 71 deaths	0.48 (0.32–0.74)	4, 6, 7, 8, 10, 13, 17, 23, 25, 26
Karavasiloglou et al. (2019) [59]	National Health and Nutrition Examination Survey (NHANES III) (USA)	F/230 (mean age = 50 years) self-reported cancer diagnosis of breast (*N* = 120), ovarian (*N* = 19), cervical (*N* = 54) or uterine cancer (*N* = 47)	Cohort study FU = 10.4 years	One 24-HR Mediterranean diet score (MDS)	High vs. low adherence MDS (1) Mortality from any cause (*N* = 121)	0.87 (0.74–1.04) for breast cancer 0.49 (0.18–1.37) for other gynecological cancers	5, 6, 7, 10, 13, 17, 20, 27, 28, 29, 30
Kenfield et al. (2014) [58]	Health Professional Follow-up Study (HPFS) (USA)	M/ 4538, diagnosed with prostate cancer, tumor stage T1–T3a, according to clinical records	Cohort study FU = 9.1 years	130-item semi-quantitative FFQ Mediterranean diet score (MDS) aMED	High vs. low adherence MDS (1) Mortality from any cause (*N* = 1181): 326 vs. 455 deaths (2) Mortality from prostate cancer (*N* = 263): 85 vs. 91 deaths	0.78 (0.67–0.90) Similar for aMED 1.01 (0.75–1.38) Similar for aMED	3, 6, 7, 9, 10, 13, 17, 20, 31, 32

F = females; M = males; FFQ = food frequency questionnaire; 24-HR = 24 h recall; CRC = colorectal cancer; NA = not available. Adjustments: 1: antiestrogen use; 2: oophorectomy status; 3: tumor stage; 4: tumor site; 5: marital status; 6: age; 7: energy intake; 8: chemotherapy; 9: treatment; 10: smoking; 11: alcohol consumption; 12: multivitamin use; 13: physical activity; 14: menopausal status; 15: parity; 16: oral contraceptive use; 17: BMI; 18: weight change since diagnosis; 19: grade of tumor; 20: time since diagnosis; 21: chemotherapy; 22: year of diagnosis; 23: sex; 24 survival time from cancer diagnosis; 25: metastases; 26: occurrence of other cancers, 27: race; 28: socioeconomic status; 29: medical conditions; 30: hormone replacement therapy; 31: Gleason score; 32: pre-diagnostic Mediterranean diet.

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
