# Peer review of "The Impact of Plant-Based Dietary Patterns on Cancer-Related Outcomes: A Rapid Review and Meta-Analysis"

_nutrients, 2020, doi:10.3390/nu12072010_

Round 1

Reviewer 1 Report

This paper is a very comprehensive summary of  the effects of plant-based diets in cancer patients. The body of work is solid and sound, despite the analysis having been done on only a limited number of papers, and its limitations are well explained.

I have two comments, which should be addressed in the Discussion:

1) The Mediterranean Diet is included in the analysis, but the Nordic Diet is not, and the authors should comment on why this was omitted.

2) There exist plant-based variations of two much-hyped diets, being the ketogenic diet and the gluten-free diet, and the authors should comment upon their possibilities.

Author Response

Reviewer 1

English language and style are fine/minor spell check required

Response: We thank the reviewer. To improve the readability, our manuscript has undergone a language review by an English native speaker (please, see track changes).

This paper is a very comprehensive summary of the effects of plant-based diets in cancer patients. The body of work is solid and sound, despite the analysis having been done on only a limited number of papers, and its limitations are well explained.

Response: We thank the reviewer for this positive feedback.

I have two comments, which should be addressed in the Discussion:

1) The Mediterranean Diet is included in the analysis, but the Nordic Diet is not, and the authors should comment on why this was omitted.

2) There exist plant-based variations of two much-hyped diets, being the ketogenic diet and the gluten-free diet, and the authors should comment upon their possibilities.

Response: The reviewer has addressed a relevant issue. We focused on the vegetarian diet and the provegetarian diet at first instance, but decide to also include the Mediterranean Diet for several reasons. This dietary pattern is by definition a plant-based diet, since it includes more than six plant-based dietary components (olive oil, nuts, fruits, vegetables, legumes and cereals). Moreover, high adherence to the Mediterranean Diet has been related to lower inflammatory and oxidative stress profiles due to its high antioxidant content. There is extensive evidence supporting the beneficial effects of this dietary pattern for cancer prevention. Thus, there are many sound arguments supporting a link between the Mediterranean Diet and cancer mortality as well.

Of course, many other healthy dietary indexes exist, including the Nordic Diet, the Dash Diet or the diet based on the Healthy Eating index. However, there are more plant-based elements in the Mediterranean Diet. For instance, the Dash Diet allows a higher consumption of meats. In the case of the Mediterranean Diet, red and processed meats come with the lowest serving recommendations per week of two or less, or at one to two servings per month. In addition, two or more servings of fish, the use of olive oil in food preparation at each meal, and a daily serving of nuts are encouraged on a Mediterranean diet plan. We therefore considered the latter as a plant-based dietary pattern, closer to the vegetarian diet, and even supplying some nutrients (iron) in higher amounts. The association between other dietary patterns and cancer mortality has been examined in other reviews; no consistent associations have been observed.

In the discussion, we have stated the following to clarify why we have focused on the Mediterranean Diet only:

Page 28, line 644: The traditional MD pattern is a well-defined plant-based dietary pattern since it is characterized by the daily use of olive oil, an abundance of plant foods such as fruit and vegetables, nuts and seeds, cereals and legumes; the consumption of fish and seafood especially in coastal regions, but moderate-to-low intake of dairy products…. Thus, the MD pattern is distinctively plant-based, and thus a valuable alternative to the vegetarian diet, as it provides a good supply of fiber, phytochemicals, vitamins and minerals, even closing some nutritional gaps of the vegetarian diet. For the reasons cited above, we considered the MD as a reference PBDP. Moreover, the MD is presumed to boost the endogenous antioxidant defense and the immune system to prevent cancer and, possibly, also fatal outcomes of this disease. With respect to cancer incidence, indeed, the MD is an established dietary pattern for cancer prevention [84].

Regarding other hyped-diets, such as the ketogenic diet and the gluten-free diet, studies regarding their association with cancer mortality are virtually inexistent. Based on the current literature, these types of diets have been related to better health outcomes (in few studies) due to their potential weight-loss and high-insulin neutralizing effects. Should these be confirmed in further studies, it would be desirable to examine their cancer mortality prevention effects too. We state this in the discussion now as follows:

Page 28, line 656: Other dietary patterns resembling plant-based diets such as the very low ketogenic diet, seem to have cancer prevention potential through weight loss and related mechanisms, but have been scarcely examined with regard to cancer mortality [85].

Reviewer 2 Report

See my comments throughout the text.

Author Response

Reviewer 2

English language and style are fine/minor spell check required 

Response: We thank the reviewer. To improve the readability, our manuscript has undergone a language review by a native speaker (please, see track changes).

This study is well presented and written. There are a few edits authors should take into account (see my comments throughout the text).

References are correct and English language is overall understandable. The paper flow is globally linear and the hypothesis is clear.

Therefore, I ACCEPT this manuscript for publication with MINOR edits.

Response: We thank the reviewer for this positive feedback and for revising the overall text. All minor edits and corrections have been incorporated into the current version of the manuscript.

In addition, we have added a reference in line 56 and line 59, as suggested by the reviewer, to support the argument on bioactive compounds in plant-based food.

Regarding line 123, we have specify why we excluded survivors of cervical lesions or colon adenomas following the reviewers suggestion

Response: We only considered invasive, malignant neoplasms in this review, to fulfill the objective. We have clarified why we excluded the above by the following statement:

Page 3, line 127: “Cancer survivors did not include survivors of cervical lesions or adenomas in the colon since these are considered benign or non-malignant lesions according to common histological classification of tumours.”

Regarding line 147, What does the definition "most recent" refer to? Authors should specify this feature in terms of years.

Response: This statement has been clarified as follows:

Page 4, line 154: “When there were several articles reporting results based on the same study population, we included the study reporting the most updated data.”

Regarding comment in line 164, “The results of all studies were presented in tabular format and summarized narratively by type of dietary pattern and cancer-related outcome. Only results with the most comprehensive adjustment for confounders were considered. When possible, results were pooled in meta-analyses”, the reviewer stated: “These inclusion/choice criteria are not precise. Authors should report numbers and ranges“. We have completed this information in the text as follows:

Page 5, line 174: To summarize these studies, we described their results and risk estimates adjusted for all potential confounders. When possible, these results were pooled in meta-analyses.

Regarding Figure 2, we think that this figure shows risk estimates in a common meta-analysis forest plot. To make Figure 1 the same, we have changed the figure layout as well. Now, we present here the weights of each study, and the pooled estimate from random effects models. Now, both figures show the forest plots in the same way.

Reviewer 3 Report

Like all meta-analyses, this work is based on other published researches, some of which may not be very reliable, and problematic because of differences in methodology, choice of populations, too few adequate researches, etc.

While it is not the authors' fault that available researches are of limited and arguable quality, even narrowing them down from around 1000 to only 24 may not have eliminated all the faulty ones.  For example, one research uses eating records of only 3 days, which is too little and unreliable, and another research included women aged 18-79, a range too wide for breast cancer discussion.

The writing seems to me to be more complicated than required, and maybe some more effort is required for making it more readable.

The conclusion of this work, namely that vegetarian diets do not reduce cancer mortality, is important as an argument against the all too common attempts to convince people to become vegetarians.

I think the work should be published, after improving its readability.

Some mistakes I noticed:

  • Line 384: "assed" should be replaced with "assessed" (a Freudian mistake :))
  • Line 399: "breast" should be replaced with "colorectal"

In conclusion, I believe it is and will be impossible to draw reliable conclusions about the effects of foods until they can be correlated with genetic tests, but this is not the authors' fault.

Author Response

Reviewer 3

I don't feel qualified to judge about the English language and style

Response: We thank the reviewer. To improve the readability, our manuscript has undergone an English language review by a native speaker (please, see track changes).

Like all meta-analyses, this work is based on other published researches, some of which may not be very reliable, and problematic because of differences in methodology, choice of populations, too few adequate researches, etc.

While it is not the authors' fault that available researches are of limited and arguable quality, even narrowing them down from around 1000 to only 24 may not have eliminated all the faulty ones.  For example, one research uses eating records of only 3 days, which is too little and unreliable, and another research included women aged 18-79, a range too wide for breast cancer discussion.

Response: The reviewer is right in highlighting these issues. We think that we should stress these arguments in the discussion as well. The following statement has been included.

Page, line 701: Dietary assessment tools (3-day records, 24-hour recalls or FFQs) also differed greatly between the studies, as well as the studied populations. For instance, there were both pre and postmenopausal breast cancer patients considered jointly in some studies.

The writing seems to me to be more complicated than required, and maybe some more effort is required for making it more readable.

Response: To improve the readability, our manuscript has undergone a language review by an English native speaker. Several statements have been rewritten along the text (please, see track changes).

The conclusion of this work, namely that vegetarian diets do not reduce cancer mortality, is important as an argument against the all too common attempts to convince people to become vegetarians.

I think the work should be published, after improving its readability.

Response: We than the reviewer for the positive feedback. The language and style has been revised by an English native speaker. We think that the readability has been improved (please, see track changes).

Some mistakes I noticed:

  • Line 384: "assed" should be replaced with "assessed" (a Freudian mistake :))
  • Line 399: "breast" should be replaced with "colorectal"

We have amended the two typos raised by the reviewer: "assed" for "assessed" in line 386 and "breast" for "colorectal" in line 401.

In conclusion, I believe it is and will be impossible to draw reliable conclusions about the effects of foods until they can be correlated with genetic tests, but this is not the authors' fault.

Response: We thank the reviewer for the comments and suggestion, which have served to improve the manuscript. As the reviewer states, research on vegetarian diet and cancer mortality is still scarce and reaching a reliable conclusion is difficult. We are far from reaching this goal. Genetics or other factors, such as the gut microbiome, together with dietary factors may help in elucidating this research question. We refer to this in the discussion/conclusion as follows:

Page 28, line 696: No study considered the interaction between dietary and genetic factors on these associations.

Page 29, line 732: Well-designed studies, considering consensus definitions of PBDPs and all pertinent factors including prognostic factors of the disease, genomics and others, are needed to determine the effect of plant-based diets on cancer survival and cancer recurrence, before and after the diagnosis of cancer